# Fitting Regularized Population Dynamics with Neural Differential Equations

**David Calhas**
Instituto Superior Técnico
INESC-ID

**Rui Henriques**
Instituto Superior Técnico
INESC-ID

## Abstract

Neural differential equations (neural DEs) are yet to see success in its application as interpretable autoencoders/descriptors, where they directly model a population of signals with the learned vector field. In this manuscript, we show that there is a threshold to which these models capture the dynamics of a population of signals produced under the same monitoring protocol. This threshold is computed by taking the derivative at each time point and analyzing the variance of its dynamics. In addition, we show that this can be tackled by projecting a highly-variant population to a lower dynamically variant space, where the model is able to capture dynamics, and similarly project the modelled signal back to the original space.

## 1 Introduction

The question we are asking is if the solution retrieved by a DE solver can be used to directly perform regression. By verifying empirically that this is not true for the current neural DE learning settings, we go further and ask if such can be done in a latent space, where the dynamics are simpler, but still informative to represent a complex population. To validate it, experiments are ran on:

- synthetic data with parametrizable complexity and variance (see Section 3);
- neuroimaging data associated with complex and noisy observations (see Section 4).
- heart rate data associated with less complex and periodic observations (see Section 5).

In particular, neural ordinary differential equations (neural ODEs) [1] are able to model signals and therefore extract vector fields that represent dynamics [2]. However, empirical observations show that neural ODEs are unable to capture dynamics of a population with a significative dynamical variance. Nonetheless, if one regularizes a latent space to have low variant dynamics, as illustrated in Figure 1, neural ODEs are able to capture that information. To this end, we introduce the concept of dynamical variance, which

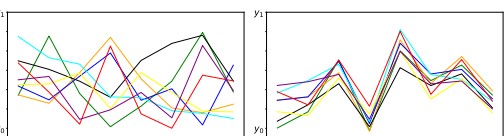

Figure 1: Variance regularization example. Consider each line to be an instance signal in a population. The plot on the left shows the population projected in its original feature space. The plot on the right shows the same population projected in a space with low dynamical variance.

refers to how much the derivative of a signal deviates from a population mean. It is hypothesized that the minimization of the dynamical variance enables the neural ODE to solve accurately a population set. The latter implies that the vectorfield of the neural ODE incorporates the dynamics of the whole population, separating instances using the initial value [3] and estimated derivatives [1, 4].

The contributions of our work are: (1) a novel loss function that allows a neural DE to learn dynamics of a population; (2) principles to reduce dynamical variance, while retaining the important dynamics of the signal, enabling the separation of different representations through the initial value and

35th Conference on Neural Information Processing Systems (NeurIPS 2021), Sydney, Australia.

estimated derivatives; (3) experiment analysis of the modelling capacity of a neural ODE under paramatrizable variance and complexity data, was performed, unravelling that it only thrives on low varying dynamical spaces and it gets better as the dynamical variance decreases; and (4) case study applications on how one can use neural DEs to capture brain and heart rate dynamics.

## 2 Methods

**Autoencoder network.** We propose the use of a neural ODE in a latent space, which is characterized by a smaller variance of dynamics relative to the original space. To this end, an encoder decoder architecture is defined, where the encoder performs the mapping between the original space, $\vec{x}$, that has representations of the raw signal, and the latent space, $\vec{z}$, that has representations with low complexity. The decoder, on the other hand, maps $\vec{z}$ to $\hat{\vec{x}}$. The encoder, $E(.|\theta_E)$, consists of an attention [5, 6] mechanism with 2 heads. The decoder, $D(.|\theta_D)$, is composed of a fully connected neural network. The encoder and decoder are parametrized by $\theta_E$ and $\theta_D$, respectively.

**Derivative estimation.** The encoder outputs a representation with the estimated derivatives for a function $f$. The derivative is given to the ODE solver by a dense neural network with parameters $\theta_f$,

$$f(\vec{z}, t|\theta_f) = \left(\frac{\hat{dz}}{dt}\right). \tag{1}$$

**Ordinary differential equations.** With the derivative given by a neural network, then a differential equation of the form $\frac{d\vec{z}_t}{dt} = f(\vec{z}_t, t)$ can be solved using the adjoint solver [7] (initial value problem solver) as done in [1]. The solution to this equation is the signal being modelled.

**Bounded complexity loss (BCL).** As for the encoder, we propose a loss that converges the dynamical complexity, $\hat{c}(\vec{z})$, of the latent representation, $\vec{z}$, to an interval defined by $[c_l, c_u]$, with $c_l, c_u \in \mathbb{R}^+$. The dynamical complexity is defined in Equation 2 and consists on taking the discrete analog of the derivative, $\Delta_t\vec{z} = \vec{z}_t - \vec{z}_{t-1}$, and subtracting with the next observed derivative, i.e. a discrete analog of the second derivative. This is done for all the observed points, $\vec{z} = \{z_1, \ldots, z_T\}$,

$$\hat{c}(\vec{z}) = \frac{1}{T-2} \sum_{t=2}^{T-1} |\Delta_t\vec{z} - \Delta_{t+1}\vec{z}|. \tag{2}$$

After computing the dynamical complexity, $\hat{c}(\vec{z})$, let $c_l, c_u \in \mathbb{R}^+ : c_l < c_u$ be predefined constants (hyperparameters), then by minimizing,

$$m(\vec{z}; c_l) = \frac{c_l}{\min(c_l, \hat{c}(\vec{z}))} - 1, \tag{3}$$

the complexity of the signal will converge to be higher than $c_l$, i.e., $\hat{c}(\vec{z}) \geq c_l$.

As the dynamical complexity of the latent representation approaches $c_l$, the penalization is higher, i.e., the gradient has a higher magnitude. However, no penalty is given if $\hat{c}(\vec{z}) \geq c_l$ and if in its turn $\hat{c}(\vec{z}) \ll c_l$, a gradient with low magnitude is computed.

The upper bound component,

$$M(\vec{z}; c_u) = \frac{\max(c_u, \hat{c}(\vec{z}))}{c_u} - 1 \tag{4}$$

has a similar behaviour as the hinge loss [8]. It gives higher penalizations than the minimum bound component for values that are distant (higher) from (than) the defined constant, $c_u$.

The two expressions, $m(\vec{z}; c_l)$ and $M(\vec{z}; c_u)$, are joined by addition in a global loss,

$$\mathcal{L}_{\text{BCL}}(\vec{z}; c_l, c_u) = m(\vec{z}; c_l) + M(\vec{z}; c_u), \tag{5}$$

formalizing the BCL loss. Its minimization implies that the complexity of the signal will converge to be between the two constants, $c_l$ and $c_u$, i.e., $\hat{c}(\vec{z}) \rightarrow [c_l, c_u]$. Please refer to the supplementary material, where you can find the plots of how these components change. The BCL loss is used to optimize $\theta_E$.

The proposed computational methodology starts with the original signal being represented at a hidden space, $E(\vec{x}; \theta_E)$, through means of a projection made by the encoder, $E$, then the latent representation of the signal, $\vec{z}$, is used to model a ordinary differential equation problem as an initial value problem. The derivative is computed using the initial value, $\vec{z}_0$, of the hidden signal representation. Using the estimated derivatives, $f(\vec{z}_0, \theta_f)$, the adjoint solver [1] models the hidden signal, $\hat{\vec{z}}$. Finally, the modelled hidden signal is transformed by a decoder to the original signal space, $D(\hat{\vec{z}}; \theta_D)$. The gradients for $\theta_E$, $\theta_D$ and $\theta_f$ are computed according to $\nabla_{\theta_E} \mathcal{L}_{\text{BCL}}(z; c_l, c_u) + ||\theta_E||_1$, $\nabla_{\theta_D} \mathcal{L}_{\text{MSE}}(\hat{\vec{x}}, \vec{x}) + ||\theta_D||_1$, $\nabla_{\theta_f} \mathcal{L}_{\text{MSE}}(\hat{\vec{z}}, \vec{z}) + ||\theta_f||_1$.

**Dynamical variance.** A metric, that measures the success of $\mathcal{L}_{\text{BCL}}$, is now described. Note that this metric computes the variance of the first derivative, while the $\mathcal{L}_{\text{BCL}}$ loss uses the second derivative. This metric focuses on the dynamical variance of the population, i.e., given the mean dynamical complexity of all instances, $\mu_{\Delta \vec{z}}$, how much do all instances deviate from it (standard deviation). Figure 1 illustrates this phenomena. Consider a population with $N$ individuals/instances. To compute the dynamical variance of the population, a different notation will be used for the complexity of an instance, $i$, which is regarded as $\Delta \vec{z}_i$. The Dynamical Variance is computed as

$$\text{Var}[\Delta \vec{z}] = \frac{1}{N} \sum_i^N (\Delta \vec{z}_i - \mu_{\Delta \vec{z}})^2. \tag{6}$$

## 3 Synthetic data

The algorithm that generates the synthetic data is explained in Appendix B.1. From Figure 2, we show that a neural ODE is unable to capture the dynamics of a population with a high variance of dynamics. It is observed that the neural ODE that operates directly on the raw signal improves as the variance, $\text{Var}[\Delta x]$, increases, which is counter intuitive. This is due to the model not adapting to the changes of the signal and ignoring them by only minimizing the error for the population set with a linear function line, similar to a linear regression. On the other hand, the neural ODE, regularized by $E$, is consistent

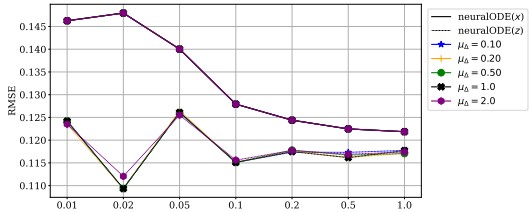

Figure 2: RMSE for the neural ODE in the latent space versus the neural ODE operating in the raw signal. In this figure, one can see the impact that the regularization of variance done by the encoder, $E$, has on modelling the true signal space.

for all the values that affect the complexity of the signal. This means the encoder performs a projection that helps modelling dynamics, which are complex enough to be retrieved from the latent to the original space representation.

## 4 Case Study: Neuroimaging data

Here we report results of the proposed methodology applied to neuroimaging data, specifically functional magnetic resonance imaging (fMRI). The feature extraction that provides the $x$-axis, $y$-axis, $z$-axis and temporal settings are described in Appendix B.2. Starting by analyz-

Table 1: Dynamical Variance comparison of the original space, $\vec{x}$, and the dynamics simplified space, $\vec{z}$.

|  | $x$-axis | $y$-axis | $z$-axis | temporal |
|---|---|---|---|---|
| $\text{Var}[\Delta \vec{x}]$ | $6.1e-3$ | $5.4e-3$ | $1.1e-2$ | $6.4e-3$ |
| $\text{Var}[\Delta \vec{z}]$ | $6.3e-4$ | $2.5e-6$ | $1.2e-3$ | $2.6e-3$ |

ing the impact of the $\mathcal{L}_{\text{BCL}}$ loss, on the $\text{Var}[\Delta \vec{x}]$, when projecting $\vec{x}$ to $\vec{z}$, $\text{Var}[\Delta \vec{z}] < \text{Var}[\Delta \vec{x}]$ for all settings. Therefore, the minimization of $\mathcal{L}_{\text{BCL}}$ equates in a lower dynamical variance in the latent space. It is also worth noting that $z$-axis and temporal settings have the highest variance in the latent space, $\vec{z}$.

RMSE and Cosine dissimilarity metrics are presented in Table 2. For all settings, the differences, of the residuals from the autoencoder (AE) [9] and latent long-short term memory network (HRNN) [10] against neuralODE($z$) [1], are all statistically significant with the maximum p-value being $4.13e - 28$. Regarding the RMSE metric, by analyzing Table 2, the neuralODE($\vec{z}$) and neuralSDE($\vec{z}$) [11] both achieved values lower than their respective peers that operated in a space where the dynamical

complexity was not regularized, namely neuralODE($\vec{x}$) and neuralSDE($\vec{x}$). The latter, validates what was already empirically proved in the previous section, using synthetic data.

The AE and HRNN baselines were outperformed with statistical significance, except for HRNN in the $y$-axis setting ($p$-value 0.1201). Their misperformance may be due to the vanishing gradient problem, because the neuralODE($\vec{z}$) and neuralSDE($\vec{z}$) both have gradients being introduced in $f$ and $E$ independently, whereas the AE and HRNN gradients for $E$ are backpropagated using the chain rule. Nonetheless, the minimization of $\mathcal{L}_{\text{BCL}}$ proved to retain information from the original space, as the decoder was able to retrieve the original signal, $\vec{x}$. It was also observed that the RMSE in the $z$-axis and temporal settings was higher than the values for $x$-axis and $y$-axis settings. This might be due to more complex dynamics being present in this dimension of the fMRI signal. Nonetheless, this also correlates with $z$-axis and temporal settings having higher dynamical variance, Var$[\Delta \vec{z}]$, in the latent space, whereas $x$-axis and $y$-axis have lower magnitudes of dynamical variance. Consequently, the minimization of $\mathcal{L}_{\text{BCL}}$ not only gives Var$[\Delta \vec{z}] < $ Var$[\Delta \vec{z}]$, but also leads to better results in a auto-encoding regression task.

Table 2: RMSE and Cosine dissimilarity metrics in the test set.

|  | RMSE | | | | Cosine dissimilarity | | | |
|---|---|---|---|---|---|---|---|---|
|  | $x$-axis | $y$-axis | $z$-axis | temporal | $x$-axis | $y$-axis | $z$-axis | temporal |
| AE | 0.1542 | 0.1528 | 0.2149 | 0.2289 | 0.3477 | 0.3894 | 0.3425 | 0.2620 |
| HRNN | 0.1464 | 0.1201 | 0.1961 | 0.2135 | 0.3254 | 0.2528 | 0.2924 | 0.2338 |
| neuralODE($\vec{x}$) | 0.1186 | 0.1279 | 0.1804 | 0.1822 | 0.2268 | 0.2920 | 0.2446 | 0.1795 |
| neuralODE($\vec{z}$) | 0.1135 | 0.1184 | 0.1633 | 0.1815 | 0.2115 | 0.2399 | 0.2068 | 0.1767 |
| neuralSDE($\vec{x}$) | 0.1202 | 0.1273 | 0.1869 | 0.1811 | 0.2273 | 0.2801 | 0.2533 | 0.1752 |
| neuralSDE($\vec{z}$) | 0.1130 | 0.1190 | 0.1631 | 0.1789 | 0.2075 | 0.2444 | 0.2172 | 0.1708 |

The Cosine Distance was computed, in order to check if the observations taken from Table 2 differ when computing a metric that looks at the pattern similarity. One observes that both metrics are in agreement. The baselines AE and HRNN perform worse than the neuralODE and neuralSDE models. The projection to the simplified latent space also improves the performance measured by this metric. Similarly, $z$-axis and temporal settings also perform worse, due to the mentioned failure of minimizing the dynamical variance.

The learned vector fields are shown in Figure 3. The $z$-axis and temporal settings have inherently more complex space, correlated with higher dynamical variance. As for the $x$-axis and $y$-axis, one observes less complex dynamics, with the $x$-axis being the space with less variability along the line.

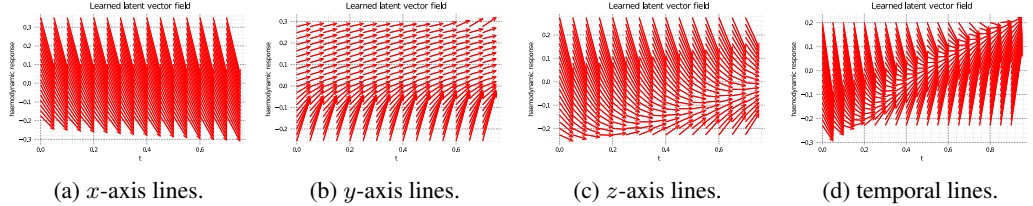

(a) $x$-axis lines.  (b) $y$-axis lines.  (c) $z$-axis lines.  (d) temporal lines.

Figure 3: Learned vector fields of each fMRI setting. Each describes the dynamics captured in the latent space, projected from the original space by $E$.

## 5 Case Study: Heart rate data

The findings collected on the MIT-BIH Arrhythmia Dataset [12] (ECG dataset described in Appendix B.3), the findings are in accordance with the ones reported in the neuroimaging case study. In this setting, the AE and HRNN yield RMSE of 0.0432 and 0.0434, respectively, competitive with the differential equation based peers, with the neuralODE($\vec{z}$) and neuralSDE($\vec{z}$) yielding an RMSE of 0.0431 and 0.0440. The performance similarity is due to the lower complexity of ECG data, being periodic and not containing significant amounts of noise artifacts. The vectorfields learned by the neuralODE($\vec{z}$) and neuralODE($\vec{x}$) show the advantage of having processed representations, $\vec{z}$, able to be represented in a vectorfield. The vectorfield of latent representations, $z$, despite not being interpretable, captures information able to be retrieved to the original representation, as is shown by the RMSE of neuralODE($\vec{x}$) and neuralSDE($\vec{x}$) being 0.0456 and 0.0458, respectively.

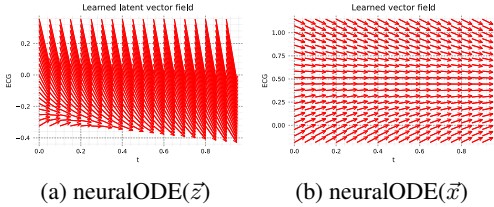

(a) neuralODE($\vec{z}$)     (b) neuralODE($\vec{x}$)

Figure 4: Learned vector fields of the neuralODE($\vec{z}$) and neuralODE($\vec{x}$).

# 6 Related Work

**Brain dynamics.** Given any system, the task of identifying its evolution/changes is known as detecting the dynamics of the system. There is a wide range of approaches that are able to analyze and model the so-claimed brain dynamics:

1. **Connectivity approaches**. As it happens with many complex systems, one way to analyze them is to model a network and apply graph theory knowledge [13]. These methods extract geometric and spatio-temporal properties from the signal. Breakthroughs have been achieved by combining these features with domain knowledge [13, 14, 15, 16, 17].

2. **Generative approaches**. More fine grained approaches are also used to study the brain, where domain knowledge is not required. Taghia et al. [18] takes advantage of the interpretability semantics of a hidden markov model (HMM) to extract critical information about the system. Huang et al. [19] model states and transitions with the purpose of detecting consciousness related dynamics. This level balances the trait of domain sepcific knowledge and technical knowledge.

3. **Regressive approaches**. Dynamics of a system can also be extracted from its raw features, by modelling it as a mathematical function [20]. This essentially depends more on machine learning stances than domain specific knowledge. However, neuroimaging signals are high dimensional and complex, limiting the feasibility of such methods to this kind of data [21].

**Neural ODEs.** Yan et al. [22] studied the robustness of neural ODEs by giving perturbed observations to the model. In contrast, we specifically target the learning of time varying dynamics by minimizing the variance of population dynamics, whereas Yan et al. [22] see neural ODEs as continuous approximations of discrete layered neural networks, one of the applications mentioned in [1]. Neural ODEs are by definition deterministic, i.e., the initial state determines the solution, yet recent advances by Li et al. [11] demonstrate that the work of Chen et al. [1] can be generalized to the stochastic version. Massaroli et al. [23] contributed to advance neural ODE research by proposing regularizers that enable the model to a faster convergence and decreased solving time. Rubanova et al. [24] used RNNs to decode the representation in a missing values/irregular observations setting.

# 7 Conclusion

Neural ODEs are not currently seen by the research community as a method capable of extracting dynamics from a population. Indeed, neural ODEs cannot fit the dynamics of populations with high dynamical variance, whereas they are able to represent low dynamical variance populations with success. In accordance, we hypothesize, that the minimization of the dynamical complexity of a space enables a neural ODE model to capture the dynamics of a population, and validate this hypothesis. A novel loss, $\mathcal{L}_{\text{BCL}}$, is introduced in order to project a highly complex dynamical space to a lower complex one. The minimization of this loss is empirically shown to also minimize the dynamical variance of a space. The encoder targeted a space that had representations with no discontinuities and low derivative variance, as inspired by the NLP technique of preserving context [25]. Furthermore, experiments were also done over a neuroimaging data (fMRI) modality, showing the possibility of using shallow neural models to capture brain dynamics.

# 8 Acknowledgments

We want to thank our colleagues at INESC-ID for the valuable input. This work was supported by national funds through Fundação para a Ciência e Tecnologia (FCT), under the Ph.D. Grant SFRH/BD/5762/2020 to David Calhas, ILU project DSAIPA/DS/0111/2018 and INESC-ID pluriannual UIDB/50021/2020.

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

# A  BCL Components

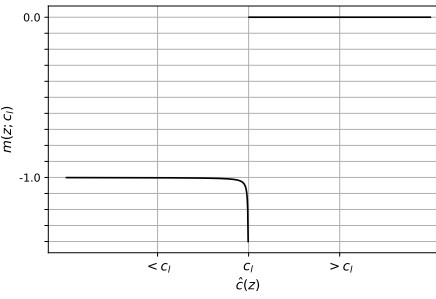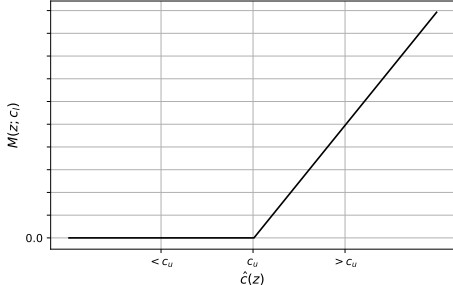

(a) Complexity bound with a lower bound, $c_l$. As the dynamical complexity of the latent representation approaches $c_l$, the penalization is higher, i.e., the gradient has a higher magnitude. No penalty is given if $\hat{c}(z) \geq c_l$ and if $\hat{c}(z) \ll c_l$, the gradient does not force linearly the representation to approach $c_l$.

(b) Complexity bound with a upper bound, $c_u$. This loss has the same behaviour as the hinge loss [8], but instead centered at $\hat{c}(z)$. This loss, $M(z; c_u)$, has higher penalizations than the minimum bound component for values that are distant from the defined constant.

Figure 5: BCL lower and upper bound components.

# B  Materials

In this Section, we provide a description of the materials used to validate the hypotheses drawn in the beginning of the manuscript. First, a description of the synthetic data is given. Then, a case study using real world neuroimaging data to extract brain dynamics is presented. Finally the validation setup is described.

## B.1  Synthetic data

To address the impact that the derivative variance has on the performance of a neural ODE, a set of synthetic lines, $l_s$, are sampled according to derivative changes $\Delta l_s \sim \mathcal{N}(\mu_\Delta, \sigma_\Delta^2)$, for all $t$ steps, and initial values $l_{s_0} \sim \mathcal{N}(\mu_0, \sigma_0^2)$, for each instance. The experiments were ran with:

- $\mu_{l_s} \in \{0.1, 0.2, 0.5, 1.0, 2.0\}$
- $\text{Var}[\Delta l_s] \in \{0.01, 0.02, 0.05, 0.1, 0.2, 0.5, 1.0\}$

A total of 10000 lines were synthesized and a split of 60% and 40% was done for training and test data, respectively.

## B.2  Case study: Neuroimaging data

In this work, to extract vector fields, two perspectives are introduced: spatial and temporal. In a spatial perspective, dynamics refer to how a signal changes through space, given a referential axis/vector. In a temporal perspective, dynamics refer to how a signal changes through time. We hypothesize that there are techniques available, due to advances in machine learning [1], which allow the direct modelling of complex systems in its raw form.

The blood flow of the brain can be captured in a functional magnetic resonance imaging (fMRI) recording session. Known as the haemodynamical activity, this modality is used to extract and uncover dynamics associated with brain functions [18, 19]. An fMRI instance $\in \mathbb{R}^{X \times Y \times Z \times T}$ is characterized by: the spatial dimension $X$ representing the x-axis voxels, the spatial dimension $Y$ representing the y-axis voxels, the spatial dimension $Z$ representing the z-axis voxels, and the temporal dimension $T$ defining the number of time points (also referred as number of volumes). All dimensions are discrete, with a fixed value used as separation/period. $X$, $Y$ and $Z$ each have a slice separation of size $v_x$, $v_y$ and $v_z$, respectively, and $T$ has a well-defined period given by Time

Response (TR) of the fMRI recording. The terms separation and period are used for spatial and temporal dimensions, respectively.

**Temporal lines extraction.** Consider a voxel characterized by the tuple $w = (w_x, w_y, w_z) \in (X, Y, Z)$, a location in space. Gathering all values of $w$ along the temporal axis provides a time series, $w$, of length $T$. To avoid the problem of individual's brain misalignment [26], a commmon sub-region of coordinates is extracted among all individuals. The coordinates were computed as a set of adjacent spirals with the center at the center point of the volume. The radius of these spirals was set to a low value, as the number of line instances increases with it. The volumes of each individual were partitioned into 30 equal sized partitions.

**Spatial lines extraction.** Consider lines that traverse a volume, a volume on time point $t$ of an fMRI is defined by three different directions: x-axis direction represented by $l_x = (w_y, w_z)$, y-axis direction represented by $l_y = (w_x, w_z)$ and z-axis direction represented by $l_z = (w_x, w_y)$. Gathering all values of $l_a$, with $a \in X \bigcup Y \bigcup Z$, creates a vector, $\vec{l_a}$, with length $S$. Lines were extracted with a fixed size of 8 voxels. From all individuals, each volume was used to extract *x*-axis, *y*-axis and *z*-axis lines. Similarly to the temporal line extraction, all lines are adjacent in its plane (either $x$, $y$ or $z$ plane), with the extracted voxels being the ones that intersect the path of the mentioned spiral, centered with the volume.

The proposed pipeline (Section 2) is tested in four different fMRI settings: temporal, *x*-axis line, *y*-axis line and *z*-axis line modelling. For the sake of completeness, each setting is subject to a hyperparameter search.

**NODDI Dataset.** The dataset [27, 28] contains 17 individuals (11 males, 6 females) with average age $32.84 \pm 8.13$ years, out of which only 10 are considered, as the remaining contain corrupted views. The simultaneous Electroencephalography-fMRI recordings were done in a resting state with eyes open setting. The subjects stayed still on a vaccum cushion during scanning. The fMRI imaging was acquired based on a T2-weighted gradient-echo EPI sequence with: 300 volumes, TR of 2160 milliseconds (ms), TE of 30 ms, 30 slices with 3.0 millimeters (mm) (1 mm gap), voxel size of $3.3 \times 3.3 \times 4.0$ mm and a field of view of $210 \times 210 \times 120$ mm. For more details please refer to [27, 28]. The dataset described is used to validate the approach proposed.For more details please refer to [27, 28]. The dataset described is used to validate the approach proposed.

### B.3 Heart rate data

Experiments were also ran on the MIT-BIH Arrhythmia Dataset [12]. The dataset contains ECG recordings of 47 individuals, sampled at at 360Hz. Each individual recording was partitioned equally in samples of $\approx 1.38$ seconds. These partitions had a total of 500 points sampled, which were downsampled to 10 by taking the maximum of a moving window. Partitions with anomalies were removed, as they contained infinities. In accordance, out of 23376 total partitions, only 22900 were considered. The latter were split with a $60/40$ ratio for train/test sets.

### B.4 Validation

For the sake of completeness, an hyperparameter search is ran with the search space defined as:

- learning rate, $lr \in \{1e-1, 1e-2, 1e-3, 1e-4, 1e-5\}$
- batch size, $b \in \{16, 32, 64, 128\}$
- activation functions , $\sigma \in \{\text{ReLU [29]}, \text{softsign [30]}, \text{hardtanh}, \text{leakyrelu [29]}, \text{mish [31]}, \text{selu [29]}, \text{swish [32]}\}$
- L1 regularization term, $\lambda \in \{1e-1, 1e-2, 1e-3, 1e-4, 1e-5, 0.0\}$
- hidden size, $h_N \in \{10, 20, 30\}$
- number of layers, $N_L \in \{1, 2, 3\}$
- augmented dimensions [33], $a \in \{0, 1, 2, 3, 4, 8, 16\}$
- upper bound of BCL, $c_u \in \{0.2, 0.5, 1.0, 1.5, 2.0\}$

The following hyperparameters are set independently for the encoder, decoder and derivative estimator networks: activation functions, L1 regularization. In addition, the hidden size and number of layers

are also set for the decoder and derivative estimator. Because the search space has a complexity of $\mathcal{O}(\Sigma^P)$, being $\Sigma$ the size of the parameter space and $P$ the number of parameters, a grid search is not feasible in time. To avoid this, a random search is implemented (it is efficient in limited time [34] and it has a high payoff for its simplicity).

In the NODDI dataset, the partitions for the train and test sets were set to $0.60$ (6 individuals) and $0.40$ (4 individuals), respectively. The hyperparameter search is ran with 200 iterations for each setting. At each iteration a 5-fold cross validation is performed in the train set and the set of hyperparameters that has the lowest average loss in the validation set of the 5 folds is chosen as the optimal hyperparameter set. The L1 regularization is optimized independently for each component (encoder, $E$, decoder, $D$,and derivative estimator, $f$). Additionally, activation function, hidden size and the number of layers are also independently optimized for $D$ and $f$. The lower bound of BCL is set to $c_l = 0.1$. For the sake of comparison, a vanilla autoencoder and an autoencoder with a recurrent layer at the latent space are implemented. The vanilla autoencoder is composed of the encoder, $E$, and the decoder, $D$. Since hyperparameters are shared with the proposed approach, the ones obtained from the hyperparameter search are recycled for these baselines. The autoencoder with a recurrent layer (LSTM [10]) at the latent space, which has the same components, $E$ and $D$, also uses the optimized hyperparameters. The neural differential equation based models are referred to as:

- neuralODE($\vec{x}$), a neural ODE operating in the original space representation, $\vec{x}$;

- neuralODE($\vec{z}$), a neural ODE operating in the latent space, $\vec{z}$;

- neuralSDE($\vec{x}$), a neural stochastic differential equation (SDE) operating in the original space representation, $\vec{x}$;

- neuralSDE($\vec{z}$), a neural SDE operating in the latent space, $\vec{z}$;

Table 3: Optimal hyperparameters obtained by the random hyperparameter search. Adam [35] was used for the training of each setting. Since the multi head attention mechanism, explained in Section 2, was used as the encoder, the activation function (linear), number of layers and hidden size are not defined for $E$.

| | $lr$ | $b$ | $\sigma_D$ | $\sigma_f$ | $\lambda_E$ | $\lambda_D$ | $\lambda_f$ | $h_{N_D}$ | $h_{N_f}$ | $N_{L_D}$ | $N_{L_f}$ | $a$ | $c_u$ |
|---|---|---|---|---|---|---|---|---|---|---|---|---|---|
| $x$-axis | 0.01 | 128 | mish | softsign | 0.0001 | $1e{-}5$ | 0.001 | 30 | 30 | 2 | 2 | 8 | 2.0 |
| $y$-axis | 0.1 | 16 | relu | relu | $1e-5$ | $1e{-}5$ | 0.001 | 30 | 10 | 3 | 3 | 8 | 1.0 |
| $z$-axis | 0.001 | 64 | leakyrelu | swish | 0.0001 | 0.0 | $1e{-}5$ | 30 | 20 | 3 | 1 | 3 | 1.5 |
| temporal | 0.001 | 128 | leakyrelu | leakyrelu | $1e-5$ | 0.0 | 0.001 | 20 | 10 | 1 | 3 | 2 | 0.5 |

