# OpenReview forum: "Fitting Regularized Population Dynamics with Neural Differential Equations"
_NeurIPS.cc/2021/Workshop/DLDE — DLDE Workshop -- NeurIPS 2021 Poster_

### Official Review · Reviewer_hmnp · 2021-10-11
**Interesting work, but a lot more explanation needed**

**Confidence:** 1

**Review:**

This work proposes and auto-encoder + neural ODE in latent space
in order to represent population dynamics.
The encoder consists of an attention mechanism with 2 heads,
and the decoder is fully connected. There is an additional cost function in latent space
which is aimed controlling what the authors call the dynamical complexity.

The introduction needs to be expanded and more literature citations are needed,
especially regarding neural ODEs and neural SDEs.
For instance, there are many methods for back-propagating through an ODE solver (see arxiv 1812.01892).

From reading the appendices and numerical sections,
I am not clear as to why the authors would like to model noisy dynamics with ODEs
instead of only focusing on SDEs. If the main goal is to learn with neural ODEs, wouldn't it be more beneficial to study/model
the noise so that its effects can be attenuated before training?

The choice of architecture for the auto-encoder also needs some explanation.
For instance, why was attention chosen to be included in the encoder?
Was there a study of various architectures to see if simpler setups lead to similar results?

An additional point of confusion is what the authors call "dynamical complexity", given by eq.(2).
I may be wrong, but it seems to be a finite difference estimate of the average second order derivative.
The authors require this value to be bounded between 2 hyper-parameters and this is achieved through the Bounded Complexity Loss.
Why give a new name to an existing concept? It may lead to more confusion in the reader.
Also, why enforce the second order derivative to be in a given, a priory, chosen range?
Shouldn't the second derivative be determined by the actual dynamics of the system?
If the noise ends up creating a huge variation in the time derivatives, then,
perhaps it would  be best to focus on filtering out some of the noise beforehand (this goes back to the previous point).
From that point of view, I guess the authors were trying to use the auto-encoder as a noise filter,
before studying the dynamics with a neural ODE, but I'm not sure whether the devised cost function aids in that purpose.

When I look at the results in table 2 and see how the neural ODE(SDE) compares between x-space and z-space,
the improvement seems to be at the percent level. One is left to wonder how the architecture complexity,
increase in number of parameters to optimize, training time etc compare with the respect to the increase in accuracy.
Perhaps a section describing this trade-off would be beneficial.

One minor suggestion would be to also include the Bayes factor in addition to the p-value in the analysis.

**Score:**

2: Borderline paper

---

### Official Review · Reviewer_Z42v · 2021-10-11
**Interesting work with potential to improve the extent to which neural ODEs capture dynamics**

**Confidence:** 1

**Review:**

The authors study how to overcome the fact that neural ODEs cannot fit the dynamics of a population with high dynamical variance. They note that in case of low dynamical variance, neural ODEs perform well, and hypothesize that minimising the dynamical complexity of a space (by using encoder-decoder architecture and projecting a space with high complexity onto a space with low complexity) can improve the fit. The authors propose a novel loss function that is empirically shown to minimize the dynamical variance of a space. Experiments are run on both synthetic data and neuroimaging data.

The main idea is presented clearly, however, I think that the paper would benefit from a slightly extended introduction, explaining the principle behind neural ODEs. Also, since the improvement in RMSE is somewhat minor in a given example, more experiments with different data and different architectures would help to further validate the hypothesis. Minor remark: in line 131, there's a typo in the inequality, Var[\Delta z] is repeated twice.

**Score:**

3: Good paper

---

### Official Review · Reviewer_8gAp · 2021-10-12
**Shows promise to improve capabilites of Neural ODEs**

**Confidence:** 1

**Review:**

The paper tries to investigate why Neural ODEs are unable to fit systems with high dynamical variance. The author(s) propose that projecting a population into another space (using an autoencoder architecture) with low dynamical variance might mitigate this problem. The author(s) introduce a novel loss function consisting of a second order finite difference term. This is used is used to train the autoencoder. Evaluations are done on one synthetic and one real-world dataset

The paper lacks clarity and explanation in some parts (Sec3). Overall idea is interesting and warrants further investigation on other datasets.

**Score:**

3: Good paper

---

### Decision · Program_Chairs · 2021-10-15

**Decision:**

Accept (Poster)

**Comment:**

Reviews were generally positive, with concerns about the clarity of the submission and some technical questions. Authors should address the reviewers’ concerns to maximize the accessibility of their poster at the workshop.